# Annihilation of exceptional points from different Dirac valleys in a 2D photonic system

M. Król [1,9], I. Septembre [2,9], P. Oliwa [1], M. Kędziora[1], K. Łempicka-Mirek[1], M. Muszyński [1], R. Mazur [3], P. Morawiak [3], W. Piecek [3], P. Kula [4], W. Bardyszewski[5], P. G. Lagoudakis[6,7], D. D. Solnyshkov [2,8]✉, G. Malpuech [2]✉, B. Piętka [1]✉ & J. Szczytko [1]✉

Topological physics relies on Hamiltonian's eigenstate singularities carrying topological charges, such as Dirac points, and – in non-Hermitian systems – exceptional points (EPs), lines or surfaces. So far, the reported non-Hermitian topological transitions were related to the creation of a pair of EPs connected by a Fermi arc out of a single Dirac point by increasing non-Hermiticity. Such EPs can annihilate by reducing non-Hermiticity. Here, we demonstrate experimentally that an increase of non-Hermiticity can lead to the annihilation of EPs issued from different Dirac points (valleys). The studied platform is a liquid crystal microcavity with voltage-controlled birefringence and TE-TM photonic spin-orbit-coupling. Non-Hermiticity is provided by polarization-dependent losses. By increasing the non-Hermiticity degree, we control the position of the EPs. After the intervalley annihilation, the system becomes free of any band singularity. Our results open the field of non-Hermitian valley-physics and illustrate connections between Hermitian topology and non-Hermitian phase transitions.

So far, topological physics has been mostly dealing with Hermitian Hamiltonians, possessing well-defined topological invariants, such as the Chern number, calculated from the eigenstates of these Hamiltonians[1]. The topological charges composing these invariants are associated with the Hamiltonian singularities in the parameter space[2], such as Dirac points. The development of non-Hermitian physics brought about new topological invariants[3], whose relation to the Hermitian ones is a particularly active topic[4].

The eigenstates of a non-Hermitian Hamiltonian are, in general, non-orthogonal. Exceptional points (EPs), where the eigenstates coalesce, can appear at the maxima of non-orthogonality when the non-Hermiticity is increased. EPs are known in optics for more than a century[5], but only recently they have been shown to allow remarkable

phenomena[6], such as specific lasing[7], unidirectional transport[8], enhanced sensing[9,10], or scattering control[11]. Their importance has been revealed thanks to their description in terms of a topological charge, characterizing a topological phase[3,12–14], which can be measured by encircling the EP in either parameter[15] or reciprocal space[4]. EPs always appear in pairs connected by a Fermi arc in the full parameter space, similar to the Weyl points in Hermitian systems, as described by the famous Nielsen-Ninomiya no-go theorem[16]. Each pair of EPs is formed from a minimum of the Hermitian coupling (e.g. band crossing).

So far, the reported non-Hermitian topological transitions were related to the creation of EPs with the increase of non-Hermiticity[4,17–20]. Typically, one Dirac point (DP) splits into two EPs by increasing non-

[1]Institute of Experimental Physics, Faculty of Physics, University of Warsaw, Warsaw, Poland. [2]Institut Pascal, PHOTON-N2, Université Clermont Auvergne, CNRS, Clermont INP, F-63000 Clermont-Ferrand, France. [3]Institute of Applied Physics, Military University of Technology, Warsaw, Poland. [4]Institute of Chemistry, Military University of Technology, Warsaw, Poland. [5]Institute of Theoretical Physics, Faculty of Physics, University of Warsaw, Warsaw, Poland. [6]Skolkovo Institute of Science and Technology, Bolshoy Boulevard 30, bld. 1, Moscow 121205, Russia. [7]Department of Physics and Astronomy, University of Southampton, Southampton SO17 1BJ, UK. [8]Institut Universitaire de France (IUF), F-75231 Paris, France. [9]These authors contributed equally: M. Król, I. Septembre. ✉e-mail: dmitry.solnyshkov@uca.fr; guillaume.malpuech@uca.fr; Barbara.Pietka@fuw.edu.pl; Jacek.Szczytko@fuw.edu.pl

Hermiticity (Fig. 1a). These EPs can then annihilate by reducing non-Hermiticity (going back to the Hermitian limit). This transition converts a Hermitian singularity into two EPs and vice versa. More complicated situations can occur when more than two levels or bands get coupled, which leads to the emergence of higher-order singularities[21,22]. These non-Hermitian transitions are based on local (geometrical) properties of states in parameter space. They do not depend on the possible presence of other DPs, and on the global geometry (topology) of eigenstates.

In this work, we demonstrate a different type of non-Hermitian topological transition in a continuous (non-periodic) 2D photonic system. We show that, if the Hermitian Hamiltonian is topologically trivial, supporting opposite-sign DPs, an EP issued from a DP can be moved towards another EP issued from another DP with which it annihilates (Fig. 1b). This process takes place upon the increase of non-Hermiticity and there is no singularity of any type (neither EPs nor DPs) left after the annihilation. It relies not only on the existence of an isolated DP, but on the global band topology, which takes into account not just one, but all singularities present in the parameter space.

## Results
### Experimental system and its model
The actual platform we study is composed of two microcavities filled with liquid crystal[23,24] (Fig. 2a). These microcavities (see Methods for structural details) host a series of photonic modes with quantized wavevectors perpendicular to the mirror plane and energies $E_N$ ($N$ is the mode number). Each mode forms a polarization doublet showing an in-plane parabolic dispersion with a 2D effective mass $m_N \sim N$. The polarization degeneracy in a doublet is lifted at all wavevectors except $k = 0$ (touching parabolas) by the splitting between TE and TM eigenmodes (Transverse-Electric and Transverse-Magnetic). This splitting acts as photonic spin-orbit coupling (SOC) characterized by a winding number 2[25,26]. The liquid crystal molecules orientation is set by an external voltage, which controls the linear birefringence $\alpha$[23,24]. A small $\alpha < (E_{N+1} - E_N)$ lifts the $k = 0$ degeneracy. The crossing between the two modes of same order (($N,N$)-case) leads to the formation of two tilted Dirac cones both carrying the same topological Berry charge $+1/2$[26]. When $\alpha$ becomes comparable with $(E_{N+1} - E_N)$, modes of different parities become energetically close and get coupled by a Rashba-Dresselhaus SOC with equal strength[23], also called emergent optical activity[27]. Here we consider $\alpha \approx (E_{N+2} - E_N)$[24], so that this optical activity is negligible. The corresponding eigenmodes look like two 2D parabola

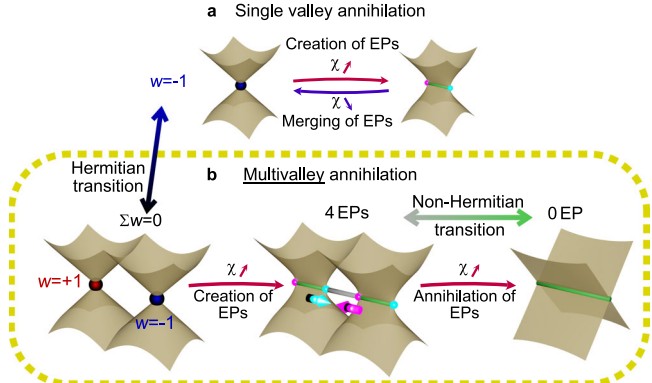

**Fig. 1 | Difference between previously considered EP annihilation and this work. a** Typical EP annihilation where only a single Dirac valley is involved. EPs are created from a DP when increasing the relative non-Hermiticity $\chi$. Conversely, they merge and form a DP when the relative non-Hermiticity decreases. **b** Annihilation of EPs described in this work, involving different valleys. 4 EPs are created from 2 DPs when increasing the relative non-Hermiticity. When it is increased further, the EPs meet and annihilate, leaving the system without any singularity. $w$ is here the winding number.

(Fig. 2b). Neglecting the losses, these two bands can be described by the following effective $2 \times 2$ Hermitian Hamiltonian written on the circular polarization basis:

$$H_{\boldsymbol{k}}^{real} = \begin{pmatrix} \frac{E_H^{N+2} + E_V^N}{2} + \frac{\hbar^2 k_x^2}{2m_x} + \frac{\hbar^2 k_y^2}{2m_y} & \Delta - \beta' k^2 - \beta(k_x - ik_y)^2 \\ \Delta - \beta' k^2 - \beta(k_x + ik_y)^2 & \frac{E_H^{N+2} + E_V^N}{2} + \frac{\hbar^2 k_x^2}{2m_x} + \frac{\hbar^2 k_y^2}{2m_y} \end{pmatrix} \quad (1)$$

where $E_H^{N+2}$ and $m_H \sim N+2$ are the energy and mass of the $N+2$th H-polarized mode, and $E_V^N$ and $m_V \sim N$ are the energy and mass of the V-polarized mode number $N$. $k_x, k_y$ are the 2D wavevector components. The spin-independent masses $m_x$ and $m_y$ are determined by the birefringence and the angle of the optical axis (see Supplementary Note I). $\beta$ is the magnitude of the TE-TM SOC, $\beta' = \hbar^2 (m_H - m_V)/4m_H m_V$ and $\Delta = (E_H^{N+2} - E_V^N)/2$. This Hermitian Hamiltonian can be written as a linear combination of identity and Pauli matrices, which defines a real effective magnetic field $\boldsymbol{\Omega}_r$ acting on the polarization pseudospin ($H = \boldsymbol{\Omega}_r \cdot \boldsymbol{S}$). The two non-zero components of the field are $\Omega_r^x = \Delta - \beta' k^2 - \beta(k_x^2 - k_y^2)$ and $\Omega_r^y = -2\beta k_x k_y$.

### Hermitian topology
This effective Hamiltonian possesses two distinct topological phases we experimentally characterize below (see also Supplementary Notes I and II for more details). If $\beta > \beta'$, the bands show two tilted DPs carrying the same topological charge (Fig. 2i), and the bands are topologically non-trivial, characterized by a non-zero winding of the pseudospin giving rise to a non-zero Chern number, if a gap is opened by breaking the time-reversal symmetry. If $\beta < \beta'$, the bands possess four tilted Dirac cones, as shown in Fig. 2c, f. Their coordinates are given by $(\pm k_{0x}, 0)$ and $(0, \pm k_{0y})$, where $k_{0,x,y} = (\Delta/(\beta' \pm \beta))^{1/2}$. The winding number of the pseudospin is $+1$ for each of the two DPs located on the $k_y$-axis, which corresponds to Berry curvature monopoles of charge $+1/2$. The DPs located on the $k_x$-axis carry a pseudospin winding number $-1$ and a Berry curvature charge $-1/2$ (Fig. 2f). The corresponding bands are therefore globally topologically trivial, with zero overall pseudospin winding and a vanishing Chern number.

Figure 3 demonstrates the above-mentioned Hermitian topological transition. It presents polarisation pseudospin (see Supplementary Note I) textures, experimentally extracted from a microcavity region with a liquid crystal layer thickness of around 3.9 μm through polarization-resolved tomography. Figure 3a is observed with 1.39 V applied to ITO (Indium Tin Oxide) electrodes. At this voltage, the modes $N + 2$ and $N$ cross each other along both wave vector directions. Four pseudospin monopoles corresponding to Dirac points are observed. The total winding of the pseudospin encircling all four Dirac points is zero, as can be seen from the high-$k$ texture (pseudospin pointing to the right). Under a different external voltage of 11 V, modes with the numbers $N$ and $N$ are almost degenerate. The pseudospin texture shown in Fig. 3b exhibits two monopoles, whose position along either the $k_x$ or $k_y$ axis is controlled by the sign of $\Delta$. In that case, the total winding number of the pseudospin is 2 (double winding stemming from the TE-TM field).

### Non-Hermitian transition
The next step is to consider the losses (line broadening), inherently present in photonic systems. Importantly, in the (N+2, N) case, these losses are significantly different for the H and V modes with $\Gamma_H = 2.04 \pm 0.04$ meV, $\Gamma_V = 1.8 \pm 0.1$ meV (see Methods and Supplementary Note IV). This requires adding a non-Hermitian part to the total Hamiltonian of the system, as already illustrated in[4,18,19]:

$$H_{\mathbf{k}} = H_{\mathbf{k}}^{real} + H^{imag} \quad (2)$$

where $H^{imag} = i(\Gamma_0 \mathbb{1}_2 + \delta\Gamma \sigma_x)$ with $\Gamma_0 = (\Gamma_H + \Gamma_V)/2$ being the mean decay rate. The term $\delta\Gamma = (\Gamma_H - \Gamma_V)/2$ defines a constant imaginary effective

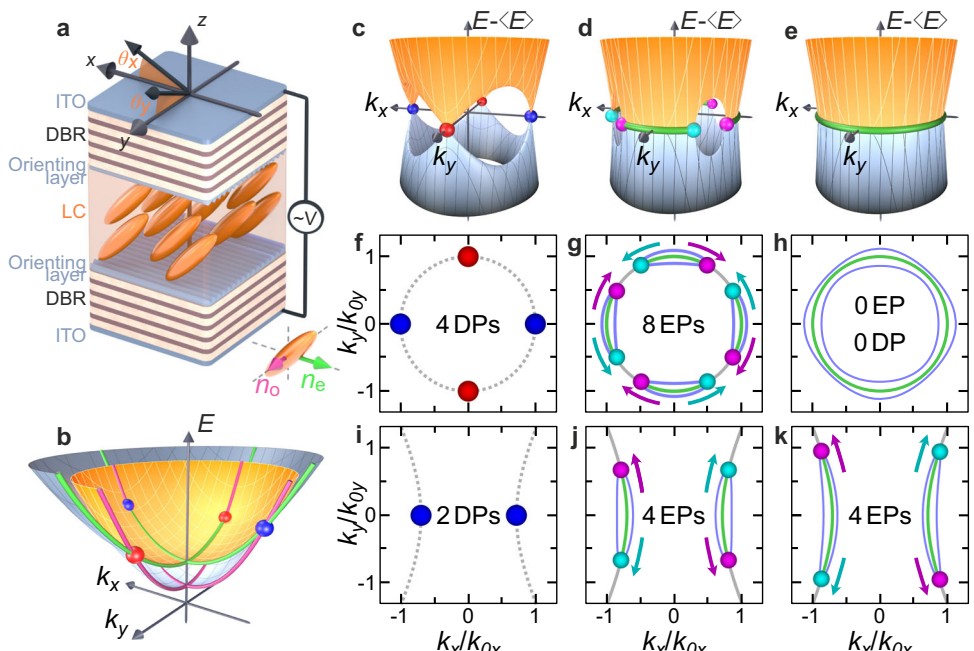

**Fig. 2 | Scheme of the experiment and the possible behaviors of the exceptional points. a** Distributed Bragg Reflector (DBR) based microcavity filled by liquid crystal (LC) molecules whose orientation is controlled by an external voltage $V$. **b** 2D Energy dispersion of the mode $N+2$ polarized H at $k=0$ and of the mode $N$ polarized V at $k=0$, $k_i = (\omega \sin\theta_i)/c$. **c–e** Difference of the real part of the energies when $\beta' > \beta$ in the Hermitian case (**c**), and non-Hermitian cases for $2\Delta = 3$ meV (**d**) and $2\Delta = 1.2$ meV (**e**). Other parameters are given in the main text. **f–h** Same as (**c–e**) but top-view. The points set the $k$-coordinate of the four DPs (**f**) and eight EPs (**g**), whereas their colors marks the sign of their topological charges. The dashed line in (**f**) shows the ellipse (appearing as a circle in these coordinates) given by Eq. (3), setting the allowed positions of EPs. EP locations are determined by the crossing of the ellipse and the blue lines given by $\Omega_r^2 - \Omega_i^2 = 0$. This crossing breaks the ellipse in Fermi arcs shown in green in (**g**, **h**) and imaginary Fermi arcs shown in gray in (**g**). **i–k** Same as (**f–g**), but when $\beta' < \beta$ and the Hermitian limit contains only two same sign DPs (**i**). EPs from different Fermi arcs (**j**, **k**) cannot annihilate, belonging to separate trajectories.

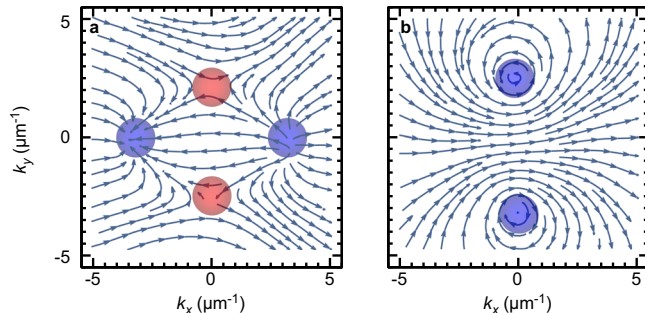

**Fig. 3 | Hermitian topological transition.** Experimental pseudospin texture in $S_1$-$S_2$ plane of lower energy band for the (**a**) $(N+2, N)$ case (4 Dirac points, zero winding) and (**b**) $(N, N)$ case (2 Dirac points, winding 2).

field along $x$: $\mathbf{\Omega}_i = (\delta\Gamma, 0, 0)^T$. As shown in Fig. 2d, g, this non-Hermitian part transforms each DP into a pair of EPs[5,19,28] connected by a line, called a Fermi arc[14], where the real parts of the eigenvalues are degenerate. The squared absolute value of the complex splitting between the eigenmodes reads $4(|\mathbf{\Omega}_r^2 - \mathbf{\Omega}_i^2|^2 + 4|\mathbf{\Omega}_r\mathbf{\Omega}_i|^2)$. The existence of an EP (zero splitting) therefore requires $\mathbf{\Omega}_r\mathbf{\Omega}_i = 0$ and $\mathbf{\Omega}_r^2 - \mathbf{\Omega}_i^2 = 0$. In our case, the first condition reads:

$$\frac{k_x^2}{k_{0x}^2} + \frac{k_y^2}{k_{0y}^2} = 1 \tag{3}$$

which determines an ellipse of possible locations for EPs (Fig. 2g, cyan and magenta points). The second condition $\mathbf{\Omega}_r^2 - \mathbf{\Omega}_i^2 = 0$ is verified along the blue curves in Fig. 2g, h. The crossing of both lines sets the coordinates of the 8 EPs:

$$k_y^e = \pm \frac{k_{0y}}{\sqrt{2}} \sqrt{1 \pm \sqrt{1 - \delta\Gamma^2/\beta^2 k_{0x}^2 k_{0y}^2}}, \tag{4}$$

$$k_x^e = \pm \frac{k_{0x}}{\sqrt{2}} \sqrt{1 \pm \sqrt{1 - \delta\Gamma^2/\beta^2 k_{0x}^2 k_{0y}^2}}. \tag{5}$$

The Fermi arc, shown in blue-green, is a part of the ellipse where $\mathbf{\Omega}_r^2 - \mathbf{\Omega}_i^2 < 0$, whereas the other part, given by $\mathbf{\Omega}_r^2 - \mathbf{\Omega}_i^2 > 0$ and shown in gray is the imaginary Fermi arc[14] with degenerate imaginary parts of the modes.

The topological charge of an EP can be defined as the winding number of the complex energy of eigenmodes around the EP[3,4,12,13]:

$$w = \frac{1}{2\pi} \oint d\mathbf{k} \cdot \mathbf{\nabla}_k \arg E_n(\mathbf{k}). \tag{6}$$

The winding numbers of the eight EPs alternate in sign along the ellipse. Increasing $\delta\Gamma$ (or decreasing $\Delta$) increases the degree of non-Hermiticity and moves the EPs away from the spawning points, along the ellipse, until they meet each other and annihilate, as shown in Fig. 2e, h. Fermi arcs connect to form a closed line of trivial degeneracy. It is not a ring of exceptional points reported in[29], because the imaginary parts of the energies are not degenerate along this whole line.

As shown above (Fig. 3), a topological transition occurs for the Hermitian part of the Hamiltonian when $\beta' = \beta$. The system switches between four DPs (globally trivial) and two DPs (non-trivial). In the case

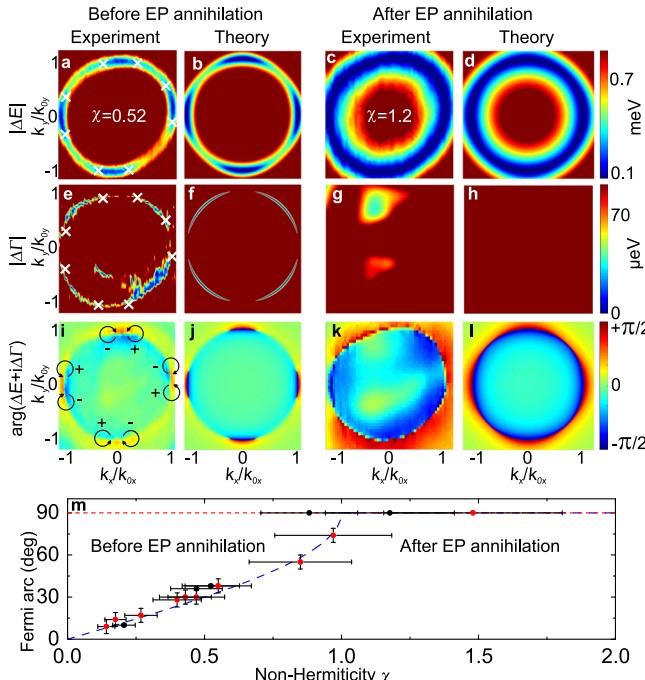

**Fig. 4 | Observation of EPs, of their topological charges and of their annihilation.** Theoretical figures are obtained by diagonalizing the Hamiltonian (2) with parameters obtained from the sample 1 and given in Methods. The procedure for the extraction of experimental energies is detailed in Methods. **a–d** Difference of the real part of the eigenenergies for $2\Delta = 3$ meV (**a, b**) and 1.2 meV (**c, d**). The color bar is saturated above 0.9 meV. The white crosses in (**a**) show the EP coordinates limiting the blue area corresponding to real Fermi arcs. **e–h** same as (**a–d**), but for imaginary part. The color bar is saturated above 90 μeV. Imaginary Fermi arcs appear in blue. **i, k** Experimental and (**j, l**) theoretical phase (argument) of the difference of complex energies. EPs are associated to a vortex phase with a phase shift $\pm \pi$ whose winding is shown by the arrows in (**i**). **m** Length of the Fermi arc with respect to non-Hermiticity. Black, red points–experiments with sample 1,2 respectively (see Methods), blue line–theory. The maximal length of each Fermi arc (90°) is marked with a thin red line. Error bars indicate the measurement uncertainty. The experimental points are obtained by varying the non-Hermiticity degree $\chi$ (controlled by the detuning $\Delta$ via applied voltage $V$).

with 2 DPs, $k_{0y}$ becomes imaginary. Equation (3) determining the location of EPs remains valid, but describes two hyperbolas (Fig. 2j, k). EPs issued from distinct DPs are moving towards infinity on separate open curves and cannot meet anymore. The EP annihilation cannot occur (See Supplementary Note VI for a detailed discussion).

Going back to the case $\beta < \beta'$, the EP annihilation occurs in Eqs. (5) when $\delta\Gamma = \beta k_{0x} k_{0y}$, which gives

$$\chi = \frac{\delta\Gamma}{\Delta} \frac{\sqrt{\beta'^2 - \beta^2}}{\beta} = 1. \tag{7}$$

The degree of non-Hermiticity $\chi$ can be changed either by increasing $\delta\Gamma$ or decreasing $\Delta$. The latter option is used in our experiment: $\Delta$ is controlled by the voltage affecting the liquid crystal molecules orientation. The experimental study of the liquid crystal cavity is performed by polarization-resolved transmission, from which we extract the real and imaginary parts of the energies of the eigenmodes and also their polarization (see Methods). The differences between the real (imaginary) part of the energies versus $k_x, k_y$ are shown in Fig. 4a, e and c,g for two values of detuning $2\Delta = 3$ meV and 1.2 meV for the cases where the EPs are present and annihilated, respectively. In Fig. 4a, e, one can observe the Fermi arcs, where the real parts of energy are degenerate and the imaginary parts split. The EPs are shown by white crosses. On the other hand, imaginary Fermi

arcs are the lines where the imaginary parts of energies are degenerate and the real parts split. The real and imaginary part of energy, and their uncertainties (see Methods and Supplementary Note V) along the Fermi arc with an EP are shown in Fig. S6. After the EPs annihilation, only a real Fermi arc remains (Fig. 4c, g). Figure 4b, d, f, h shows the corresponding theoretical results, obtained using the effective non-Hermitian Hamiltonian (2). The agreement between experiment and theory is excellent. We note that in ref. 24, the very same structure was studied in the regime of crossing between the $N + 2$ and $N$ modes. The key difference is that the experiment was performed at higher detuning $\Delta$, so with a very small non-Hermiticity degree. This is also the regime for which Fig. 3 was measured. In that case Fermi arcs are very small, the EPs could not be resolved and an Hermitian description of bands is appropriate.

The topological charge measurement is presented in Fig. 4i–l. Fig. 4i, k show a map of the phase of the complex energy of the lower mode for $2\Delta = 3$ meV and $2\Delta = 1.2$ meV, as previously. In both cases, the real Fermi arcs appear as a sharp phase shift. Very clear phase vortices are visible at the EP positions in Fig. 4i and are absent in Fig. 4k. These features are in excellent agreement with the simulations based on the effective non-Hermitian Hamiltonian (2), shown in Fig. 4j, l. The topological charge of each EP $w = \pm 1/2$ is determined by the direction of the phase vortex winding. These charges are opposite for the EPs originating from different DPs, which ultimately allows their annihilation.

Figure 4m demonstrates the control of the Fermi arc angular size related to the EPs position through the experimental tuning of the non-Hermiticity $\chi$. The topological transition associated with the EP annihilation is clearly visible taking place for $\chi = 1$. To demonstrate that the observed behavior does not depend on a particular sample, we have performed extra measurements with a sample characterized by different parameters. Black and red dots in panel 4m correspond to samples 1 and 2 respectively, whereas all data shown in panels 4(a–l) are from sample 1. The theoretical curve is universal, it does not have any fitting parameters.

## Discussion

Non-Hermitian transitions in two-band systems through EP merging are typically related to a single Hermitian singularity[3,13,14,18]. Here, we consider the merging of EPs originating from different DPs upon increasing the non-Hermiticity, which to our knowledge was not reported before. It can be viewed as a first example of non-Hermitian multivalley physics and demonstrates the link between Hermitian topology and non-Hermitian phase transitions. Indeed, there is no singularity of any type after the transition, as in the transition with the annihilation of 2D DPs carrying opposite charges[30]. The phase transition we observe could be realized in other multivalley systems, like artificial graphene with a $\sigma_x$ non-Hermitian contribution.

From an applied perspective, our work sets microcavities along-side the waveguide-based photonic systems[31] as a reconfigurable platform for exploring non-Hermitian topology. We demonstrate the tuning of the EP coordinates in k-space by simple modification of an external voltage, in a micro-device, at optical frequencies. This could allow to control the angle of emission of the modes surrounding the EPs, which are known to possess remarkable properties[6]. Another interest of the planar cavity platform is that it allows implementing interacting photons modes (exciton-polaritons)[4] possibly up to the recently demonstrated single-photon non-linearity[32]. These possibilities combined with our present finding could allow to address in future non-Hermitian topological physics for strongly interacting-correlated particles.

## Methods
### Samples
Both cavities consist of two distributed Bragg reflectors made of 6 SiO$_2$/TiO$_2$ pairs with maximum reflectance at 550 nm grown on glass

plates with ITO electrodes. Space between the DBRs is filled with highly birefringent liquid crystal with $\Delta n = 0.41$. To obtain homogeneous orientation of LC, both DBRs are finished with structured polymer orienting layer. The cavity 1 studied in Figs. 4, (S4–S7) has a total thickness of the LC layer of approx. 1.8 μm. The cavity 2 studied in Figs. 3, 4m and Fig. S8 has a thickness of 3.2 μm.

## Experimental setup

Experimental results were obtained in transmission configuration. Broadband light from a LED diode was focused on the sample with a microscope objective with $50 \times$ magnification and numerical aperture $NA = 0.6$. Transmitted light was collected by another objective with $20 \times$ magnification and $NA = 0.4$. Fourier plane of the collecting objective was imaged on the entrance slit of a monochromator equipped with a CCD camera. Both wave vector directions were measured by scanning of the image across the slit by the automated movement of the imaging lens. Data was collected independently for 6 incident light polarizations; linear: horizontal, vertical, diagonal, anti-diagonal and circular $\sigma^+$, $\sigma^-$ by adjusting angles of half wave plate and quarter wave plate after fixed linear polarizer. LC layer anisotropy in $x-z$ plane was controlled by external square waveform applied to ITO electrodes with 1 kHz frequency and amplitude of 1.77 V (Fig. 4a, e, i) and 1.72 V (Fig. 4c, g, k).

## Linewidth extraction

We extract the real and imaginary parts of the energies (that is, the positions and the linewidths) of the modes from the polarization-resolved spectra by fitting them with the Voigt function, in order to account both for the homogeneous broadening due to the mode lifetime and for the inhomogeneous broadening due to disorder. Only the homogeneous part of the broadening (Lorentzian linewidth) can give rise to non-Hermiticity and is accounted for by the Hamiltonian (2). Parallel computing is used to speed up the extraction for the whole reciprocal space with high resolution. An example of the energy spectrum in two polarizations, together with its fit, is provided in the Supplementary Note IV.

The variation of the difference of the linewidths $\delta\Gamma$ with detuning $\Delta$ is negligible, because the linewidths scale as the energies of the modes and their difference scales as the birefringence $\Delta n$, responsible for the difference of the energies between the modes of the same order $E_H^N - E_V^N$, whereas the detuning $\Delta$ between the almost degenerate modes $E_H^{N+2} - E_V^N$ does not account for the whole bire-fringence, but only for a small part of it. Indeed, in our case the overall splitting due to the birefringence $E_H^N - E_V^N$ ~ 200 meV, and its change with voltage (corresponding to the variation of $\Delta$) is of the order of 1 meV. The variation of $\delta\Gamma$ with voltage is of the same order as that of the birefringence (about 1%), which is smaller than the experimental uncertainty of this parameter (20%).

The uncertainty on the linewidth is the main source of uncertainty on the non-Hermiticity degree $\chi$, used to plot Fig. 4m. Other non-negligible contributions come from the uncertainty on the polarization splittings $\beta$ and $\beta'$. As to the length of the Fermi arcs, we have used 3 arcs out of 4 for averaging for sample 1, since the 4th one (the one at $k_y < 0$) is systematically affected by experimental measurement problems. The lengths of the remaining 3 arcs are very close (e.g. 36, 37, and 38 degrees for the experiment presented in Fig. 4a). For sample 2, we have used all 4 arcs, and the uncertainty is slightly larger. The uncertainty on the position of the EPs is relatively low, because this position is constrained by 2 independent measurements and a third one combining them (real and imaginary parts of the energy, and its phase).

## Experimental uncertainty and the evidence for topological transition

The 2D images shown in Fig. 4 of the main text do not allow to indicate the uncertainties, which are important to prove that the transition

associated with the annihilation of the EPs indeed takes place. To show this, we plot in Fig. S6 the real and imaginary part of energy, and their uncertainties along the Fermi arc when an EP is present. We also show in Fig S7 a cross-section of Fig. 4a, d of the main text near the top right corner of the Fermi arc denoted by $k'$ to clearly show the difference between the case where EPs exist and when they annihilate. In one case, this cross-section crosses an imaginary Fermi arc, and in the other case – a real Fermi arc that forms a full circle.

## Parameters of the Hamiltonian

The theoretical panels of Fig. 4 were calculated using the following parameters, obtained from the sample 1 by fitting the dispersion (Supplementary Note III) and from the linewidth extraction discussed above: $m_x = (1.34 \pm 0.07) \times 10^{-5} m_0$, $m_y = (1.08 \pm 0.06) \times 10^{-5} m_0$, $\beta = 0.080 \pm 0.03$ meV$\mu$m$^2$, $\beta' = 0.47 \pm 0.03$ meV$\mu$m$^2$. Other parameters were already given in the text, but we provide them here for convenience: $2\Delta = 2.7 \pm 0.1$ and $1.2 \pm 0.1$ meV, $2\delta\Gamma = 0.24 \pm 0.05$ meV.

The parameters of the sample 2 are given here for comparison: $m_x = (1.27 \pm 0.03) \times 10^{-5} m_0$, $m_y = (1.14 \pm 0.03) \times 10^{-5} m_0$, $\beta = 0.23 \pm 0.01$ meV$\mu$m$^2$, $\beta' = 0.35 \pm 0.01$ meV$\mu$m$^2$, $2\delta\Gamma = 0.5 \pm 0.07$ meV.

## Data availability

The data generated in this study are available in the Open Science Framework (OSF) repository: https://osf.io/jnx8k/?view_only=16426f 9a35404264badaaa93162060a7.

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

## Acknowledgements

We acknowledge useful discussions with C. Leblanc. This work was supported by the National Science Centre grants 2019/35/B/ST3/04147, 2019/33/B/ST5/02658, 2018/31/N/ST3/03046 and 2017/27/B/ST3/00271, and the Ministry of National Defense Republic of Poland Program – Research Grant MUT Project 13-995 and MUT University grant (UGB) for the Laboratory of Crystals Physics and Technology for year 2021, and the European Union's Horizon 2020 program, through a FET Open research and innovation action under the grant agreements No. 899141 (PoLLoC) and No. 964770 (TopoLight). We also acknowledge the support of the ANR Labex Ganex (ANR-11-LABX-0014), and of the ANR program "Investissements d'Avenir" through the IDEX-ISITE initiative 16-IDEX-0001 (CAP 20-25). P.G.L. acknowledges the support of the UK's Engineering and Physical Sciences Research Council (grant EP/M025330/1 on Hybrid Polaritonics), the support of the RFBR project No. 20-52-12026 (jointly with DFG) and No. 20-02-00919.

## Author contributions

J.S., G.M., D.D.S., W.P., B.P., and P.G.L. acquired funding; M.Kr., P.O., M.Kę., M.M. and K.Ł-M. performed the experiments under the guidance of B.P. and J.S. with strong support from P.G.L.; P.K. synthesized liquid crystal; R.Ma., P.M., and W.P. constructed and fabricated the LC microcavity; I.S. performed theoretical analysis under the guidance of G.M. and D.D.S.; G.M. and D.D.S. proposed the Hamiltonian of the system with the non-Hermitian part; M.Kr. and W.B. developed numerical Berreman method modeling. All authors participated in the interpretation of experimental data; J.S., D.D.S., G.M., and W.P. supervised the project; I.S., M.Kr., G.M., D.D.S. wrote the manuscript with input from all other authors. M.Kr., B.P., and I.S. proposed the visualization of experimental and theoretical data.

## Competing interests

The authors declare no competing interests.
