## [Peer Review File · Nature Communications]

REVIEWERS' COMMENTS

Reviewer #1 (Remarks to the Author):

After reading the response and the revised manuscript, I believe the present version addressed my question and comments, and this work could be accepted in Nature Communications.

Small error appears at page 8: “the total winding of the pseudo encircling” should be “the total winding of the pseudo spin encircling”?

Reviewer #2 (Remarks to the Author):

The paper is suitable for publication in Nature Communications as is.

Response to reviewers' comments

Below, we provide a point-by-point response to the comments of the reviewers.

Reviewer #1 (Remarks to the Author):

Reviewer writes:

After reading the response and the revised manuscript, I believe the present version addressed my question and comments, and this work could be accepted in Nature Communications.

Small error appears at page 8: "the total winding of the pseudo encircling" should be "the total winding of the pseudo spin encircling"?

Authors reply:

We thank the reviewer for the positive appreciation of our efforts in our previous response, and for the recommendation to accept the manuscript. We also thank the reviewer for spotting the misprint, indeed, the "pseudo" meant "pseudospin", as suggested by the reviewer.

Reviewer #2 (Remarks to the Author):

Reviewer writes:

The paper is suitable for publication in Nature Communications as is.

Authors reply:

We thank the reviewer for the recommendation to accept our manuscript to Nature Communications.

Sincerely yours,

The authors